# Whole-genome sequencing reveals origin and evolution of influenza A(H1N1)pdm09 viruses in Lincang, China, from 2014 to 2018

Xiao-Nan Zhao[1]☉, Han-Ju Zhang[2]☉, Duo Li[1], Jie-Nan Zhou[1], Yao-Yao Chen[1], Yan-Hong Sun[1], Adeniyi C. Adeola[3], Xiao-Qing Fu[1], Yong Shao[3]*, Mei-Ling Zhang🄳[1]*

**1** Department of Acute Infectious Diseases Control and Prevention, Yunnan Provincial Center for Disease Control and Prevention, Kunming, Yunnan, China, **2** Lincang Center for Disease Control and Prevention, Lincang, Yunnan, China, **3** State Key Laboratory of Genetic Resources and Evolution, Chinese Academy of Sciences, Kunming Institute of Zoology, Kunming, Yunnan, China

☉ These authors contributed equally to this work.
* meilingz2011@163.com (MLZ); Yong_Shao_dws@126.com (YS)

**Data Availability Statement:** The 45 genome sequencing data of A(H1N1)pdm09 strains have

## Abstract

The continuous variation of the seasonal influenza viruses, particularly A(H1N1)pdm09, persistently threatens human life and health around the world. In local areas of southwest china, the large time-scale genomic research on A(H1N1)pdm09 is still insufficient. Here, we sequenced 45 whole-genome sequences of influenza A(H1N1)pdm09 viruses in Lincang, China, from 2014 to 2018, by next-generation sequencing technology to characterize molecular mechanisms of their origin and evolution. Our phylogenetic analyses suggest that the A(H1N1)pdm09 strains circulating in Lincang belong to clade 6B and the subclade 6B.1A predominates in 2018. Further, the strains in 2018 possess elevated evolutionary rate as compared to strains in other years. Several newly emerged mutations for HA (hemagglutinin) in 2018 are revealed (i.e., S183P and R221K). Intriguingly, the substitution R221K falls into the RBS (receptor binding site) of HA protein, which could affect antigenic properties of influenza A(H1N1)pdm09 viruses, and another substitution S183P near to RBS with a high covering frequency (11/14 strains) in 2018 is exactly located at the epitope B. Notably, the NA (neuraminidase) protein harbors a new mutation I23T, potentially involved in N-glycosylation. Based on the background with a higher evolutionary rate in 2018 strains, we deeply evaluate the potential vaccine efficacy against Lincang strains and discover a substantive decline of the vaccine efficacy in 2018. Our analyses reaffirm that the real-time molecular surveillance and timely updated vaccine strains for prevention and control of influenza A(H1N1)pdm09 are crucial in the future.

## Introduction

The outbreak of influenza A(H1N1)pdm09 pandemic in 2009–2010 has resulted in a large amount of illness or death globally [1–4]. It has become a serious worldwide public health problem. Since the influenza A(H1N1)pdm09 pandemic during 2009–2010, the activity of

been deposited at NCBI Gene Bank Database with accession codes: MT540604-MT540963.

**Funding:** The authors received no specific funding for this work.

**Competing interests:** The authors have declared that no competing interests exist.

seasonal influenza viruses has relatively inhibited and further sustained a low level in China until the national epidemic during Sep 2017 and Feb 2018 with elevated influenza-related hospitalization, illness and death [5, 6]. Meanwhile, of in-hospital mortality during 2017–2018, the vast majority of infections were caused by influenza B and A/H1N1 [5, 6].

It is well-known that the influenza virus genome consists of eight gene segments and their high variability originates from sequence changes located in eight gene segments (HA, NA, M, NP, NS, PA, PB1 and PB2) [7]. The HA gene encodes the viral surface glycoprotein HA, which is responsible for binding to sialic acids (SAs), serving as the viral receptors on host cells and for fusion of the viral and host cell membranes on endocytosis. Another gene NA encodes a neuraminidase helping viruses to cleave SAs from host cells and virus particles [8, 9]. Afterwards, the HA is cleaved by host proteases into HA1 and HA2 subunits. The HA1 subunit harbors the globular head domain containing five distinct antigenic epitopes (Sa, Sb, Ca1, Ca2, and Cb) and the sialic acid receptor-binding sites (RBSs) [10, 11]; The HA2 subunit harbors the stalk domain which induces fusion between the viral envelope and host endosomal membrane [12]. Consequently, mutations within these domains might have potential effectiveness on the attachment of the virus to host cells and the recognition of the virus by the neutralizing antibodies aroused in human [13, 14]. Influenza viruses can utilize the efficient antigenic drift to escape from the elimination of the host's immune system. Thus, WHO needs to update components of the vaccine strains annually to ensure maximum protection [14]. More importantly, the evaluation of the vaccine efficacy is essential for the selection of high-quality vaccine strains each year [15, 16].

The rapid evolution of genome sequences of seasonal influenza viruses poses an underlying challenge for vaccine development and selection. Therefore, in the present study, we aim at elucidating molecular mechanisms of origin and genetic variability of influenza A(H1N1) pdm09 viruses in Lincang, China, from 2014 to 2018 and further providing real-time monitoring resources for the development of effective vaccine, which is particularly vital for prevention and control of seasonal influenza.

## Materials and methods

### Ethics consideration

The project entitled "Whole-genome sequencing reveals origin and evolution of influenza A (H1N1)pdm09 viruses in Lincang, China, from 2014 to 2018", submitted by investigator, Mei-Ling Zhang, Department of acute infectious disease control and prevention, has been approved by the meeting of ethics committee of Yunnan Provincial Centers for Disease Control and Prevention, according to Chinese ethics laws and regulations, with approval number YNCDC2014001. The ethics committee has approved the oral consent procedure to protect patients' privacy including minor patients, whose oral consent originated from their parents. It is recognized that the right and the welfare of the subject are adequately protected.

### Specimen collection and isolation and identification of virus

The surveillance was performed in patients with an influenza-like illness (ILI) who were admitted to Lincang people's hospital in Lincang, China, from January 1, 2014 to December 31, 2018. The ILI was defined by unexpected high fever ≥38˚C with respiratory symptoms (i.e., cough and sore throat). The nasopharyngeal swabs were placed using the viral transport media and sent to the Lincang Center for Disease Control and Prevention within 24 hours for further detection of the influenza viruses. The collected samples were screened for influenza virus by RNA extraction and the real-time reverse transcription-polymerase chain reaction (RT-PCR). The positive samples were cultured in the Madin–Darby canine kidney (MDCK)

cells for 4–7 days to isolate influenza viruses, which were later confirmed by haemagglutina-tion (HA) assay. The influenza subtypes were identified by hemagglutination inhibition (HI) assay with post-infection ferret antisera raised against vaccine strains (A/California/7/2009 (H1N1)pdm09 (2009–2016) and A/Michigan/45/2015 (2017–2018)) and circulating strains from the Chinese National Influenza Center. And specific experimental procedures comply with Technical Guidelines for National Influenza Surveillance (2017 Edition) (http://www.chinaivdc.cn/cnic/zyzx/jcfa/201709/t20170930_153976.htm). The identified results were reviewed by Yunnan Center for Disease Control and Prevention and then sent to the Chinese National Influenza Center for confirmation.

## Whole genome sequencing of influenza A(H1N1)pdm09

Viral nucleic acids of isolated strains were extracted using the QIAamp Viral RNA Mini Kit (Qiagen, Hilden, Germany). They were subsequently applied for cDNA synthesis and PCR amplification by SuperScript$^{TM}$ III One-Step RT-PCR System with Platinum$^{TM}$ *Taq* High Fidelity. The amplification primers were Uni-12/Inf1 (primer A): 5'-GGGGGGAGCAAAAGC AGG-3', Uni-12/Inf3 (primer B): 5'-GGGGGGAGCGAAAGCAGG-3' and Uni-13/Inf1 (primer C): 5'-CGGGTTATTAGTAGAAACAAGG-3'. Then, the amplification products were purified and quantified by QIAquick 96 PCR Purification Kit and Qubit$^{TM}$ dsDNA HS Assay Kit. Next, we performed a series of procedures such as tagment genomic DNA, amplify librar-ies, clean up libraries, normalize libraries and dilute libraries to final loading concentration by Nextera XT DNA Library Prep kit referring to the kit instructions. In the end, we thawed reagent cartridge (MiSeq v2 Reagent Tray 300 cycles-PE), loaded the pooled libraries onto the reagent cartridge in the designated reservoir and operated MiSeq sequencer applied to sequencing according to the illumine MiSeq system guide. The raw sequencing paired-end reads were imported to CLC Genomics Workbench 9.5.2 (Qiagen, Denmark) and further were trimmed and performed for *de novo* assembly into contigs. The contigs were imported to National Center for Biotechnology Information (NCBI) for blast (https://www.ncbi.nlm.nih.gov/genomes/FLU/Database/nph-select.cgi?go=database), and the best-hit reference strains were selected for mapping to produce consensuses. The 45 genome sequencing data of A (H1N1)pdm09 strains have been deposited at NCBI Gene Bank Database with accession codes: MT540604-MT540963.

## Construction of phylogenetic trees and prediction of glycosylation sites

The 14 global reference strains including three northern hemisphere vaccine strains (2014–2020) recommended by WHO were retrieved from the Global Initiative on Sharing All Influ-enza Data (GISAID) database (http://www.gisaid.org) (S1 Table). Multiple sequence alignment was guided by ClustalW 1.82 (http://www.clustal.org/). Phylogenetic analyses for the eight gene segments were executed by the MrBayes v3.2.1 software under HKY+I+G nucleotide sub-stitution model [17], and further we presented and annotated our phylogenies by using Fig-Tree v1.4.3 (https://mac.softpedia.com/get/Graphics/FigTree.shtml). The chain length was set to 10,000,000, with the first 1,000 samples burned in and other parameters were regarded as defaults.

Meanwhile, we utilized the MCMC algorithm of BEAST v1.10.4 (http://tree.bio.ed.ac.uk/software/beast/) to evaluate the most recent common ancestor (TMRCA) from our isolated strains. In detail, HKY substitution model and uncorrelated relaxed clock with lognormal relaxed distribution were used in our analysis, and the 10,000,000 MCMC chain length was performed. The Tracer v1.5 (http://tree.bio.ed.ac.uk/software/tracer/) was used to detect the convergence. Furthermore we used the TreeAnnotator v1.10.4 software of the BEAST v1.10.4

to annotate the target Maximum clade credibility tree (MCCTREE) with 10% Burnin as the number of trees. Additionally, we also used FigTree v1.4.3 to visualize our MCCtree. Divergence time of ancestral nodes with 95% highest density probability (HPD) and mutation rates were estimated by BEAST v1.10.4, respectively. The numbering system of HA amino acid was applied after removing the signal peptide. Potential N-linked glycosylation sites were predicted using the NetNGlyc 2.0 web server (http://www.cbs.dtu.dk/services/NetNGlyc) with a threshold value of >0.5.

## 3D model visualization of HA and NA protein

The 3D structures of HA and NA proteins were predicted using the SWISS-MODEL web server (https://www.swissmodel.expasy.org/), and the identified mutations within antigenic sites and other important sites were marked with PyMOL (https://www.pymol.org/). The relative amino acid frequency in the epitope of HA1 was obtained using WebLogo [18].

## Potential mutants contributing to functional changes

In order to further elucidate functional mutations, we used PROVEAN (http://provean.jcvi.org) to predict potentially functional impact of amino acid substitutions for HA and NA protein. The software provides the conservation degree of amino acids relative to other published sequences and further employs a score to define the potential effect of the substitution on protein function. A default cutoff score of less than −2.5 indicates a high probability of functional mutation [19].

## Estimation of vaccine efficacy of influenza A(H1N1)pdm09 strains

The vaccine efficacy of A(H1N1)pdm09 was estimated using the $P_{epitope}$ method, which is expressed as the proportion of amino acid substitutions in the dominant HA epitope [15]. To determine antigenic distance between the vaccine strain and the circulating strains, we took into account five epitopes (A-E) of the A(H1N1)pdm09 strains in analogy to the A(H3N2) virus [20]. The $P_{epitope}$ is calculated by dividing the number of amino acid substitutions in HA1 epitope by the total number of amino acids in the epitope. This association between vaccine efficacy and $P_{epitope}$ is given by $E = (0.53–1.19 \times P_{epitope}) \times 100$ for influenza A(H1N1) pdm09 virus [16, 21, 22].

## Results

### Isolation rate, demographic characteristic and sampling distribution of influenza

Of the 6,767 influenza-like illness samples collected in Lincang people's hospital, during the study period (January 1, 2014 to December 31, 2018), 392 strains were isolated from the samples of influenza-like cases. Among that, the isolation rates of influenza A and B strains were 3.04% and 2.75%, respectively (S2 Table). Among the 392 cases infected with influenza viruses, male accounted for the majority (61.73%), mainly in the age group under 5 years old, and scattered children were the main infected population (S3 Table). 45 strains were randomly selected from total 112 isolated influenza A(H1N1)pdm09 strains in Lincang, China to perform whole-genome sequencing. The sampling distribution of selected strains was plotted in Fig 1. The number of sequencing strains covered ~44.3% of the total A(H1N1)pdm09 strains per year on average. Because none of A(H1N1)pdm09 strains was isolated in 2015, this year was deleted in our downstream analyses.

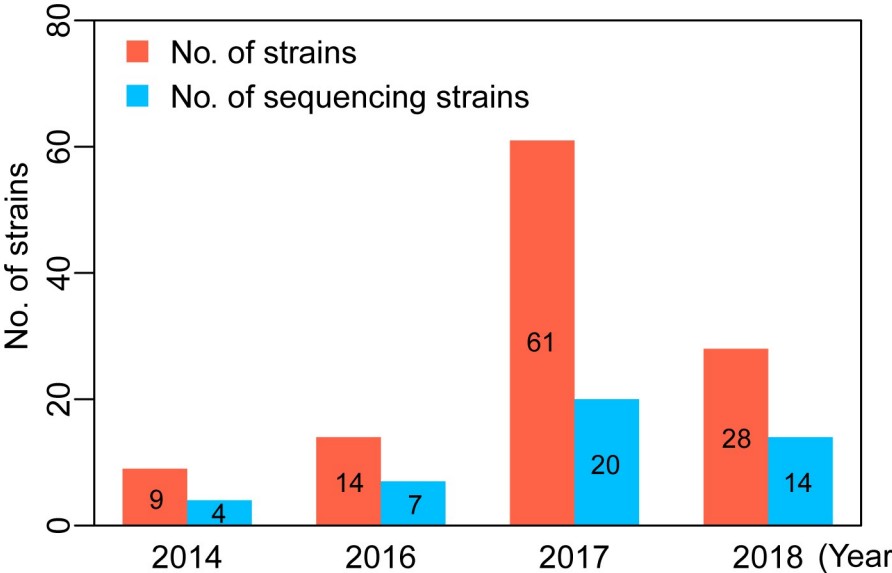

**Fig 1. Sample distribution of total and sequencing strains of influenza A(H1N1)pdm09 viruses in Lincang, China, from 2014 to 2018.**

## Phylogeny and mutation analysis of HA of influenza A(H1N1)pdm09 strains

Here, we integrated phylogenetic and antigenic epitope analyses to understand potential evolutionary mechanisms of influenza A(H1N1)pdm09 viruses in Lincang, China, from 2014 to 2018. Mutations occurred in antigenic epitopes of A(H1N1)pdm09 viruses can potentially influence their pathogenicity [23]. Based on the sequence homology, mutations observed in five epitopes (Sa, Sb, Ca1, Ca2, and Cb) were mapped to the H3 epitopes (B+D, B, C+D, A+D and E), respectively (S4 Table). This was an alternative way to identify antigenic sites in A (H1N1)pdm09 viruses as the previous study [20, 24].

According to the Mrbayes phylogenetic tree of HA genes (Fig 2 and S1 Fig), all the 45 strains characterized by cluster 6B (A/South Africa/3626/2013-like) showed representative amino acid substitutions, including P83S (epitope E), D97N, K163Q (epitope D), S185T (epitope B), S203T, K283E (epitope C), I321V, E374K, S451N and E499K (S4 Table).

Furthermore, the 6B clade grouped into two distinct cluster lineages, represented by A/ Michigan/45/2015 (6B.1) and A/Iowa/53/2015 (6B.2). The 2014 strains and most (4/7) of 2016 strains located in 6B.2 lineage with differentiable amino acid substitutions, such as V152T (epitope B), V173I (epitope D), A261S (epitope E), E491G and D501E. All 2017–2018 strains and residual three 2016 strains were clustered into 6B.1 lineage, which are widespread throughout the world since 2016, harboring substitutions S84N (epitope E), I216T (epitope D) and S162N (epitope B), representing a potential N-glycosylation site (S4 Table).

Moreover, two 2016 and four 2017 strains clustered with A/Victoria/55/2017 shared an additional substitution A215G (epitope D), and most (15/20) strains from 2017 shared extra substitutions I324V and K454R. And most (13/14) 2018 strains clustered into the 6B.1A subgroup (A/Brisbane/02/2018-like) with mutations S74R (12/13)/S74K (1/13) (epitope E), I96V (9/13) (epitope D), S164T (13/13) (epitope D), S183P (11/13) (epitope B), I295V (13/13) (epitope C) and A256I (10/13) (S4 Table). Additionally, one 2018 strain (A/Yunnan-Linxiang/ SWL1765/2018) has a substitution R221K occurred at the HA RBSs (positions 184–191, 218–225, 131–135, Y91, W150, H180 and Y192) [25]. The substitution S185T also overlapped with

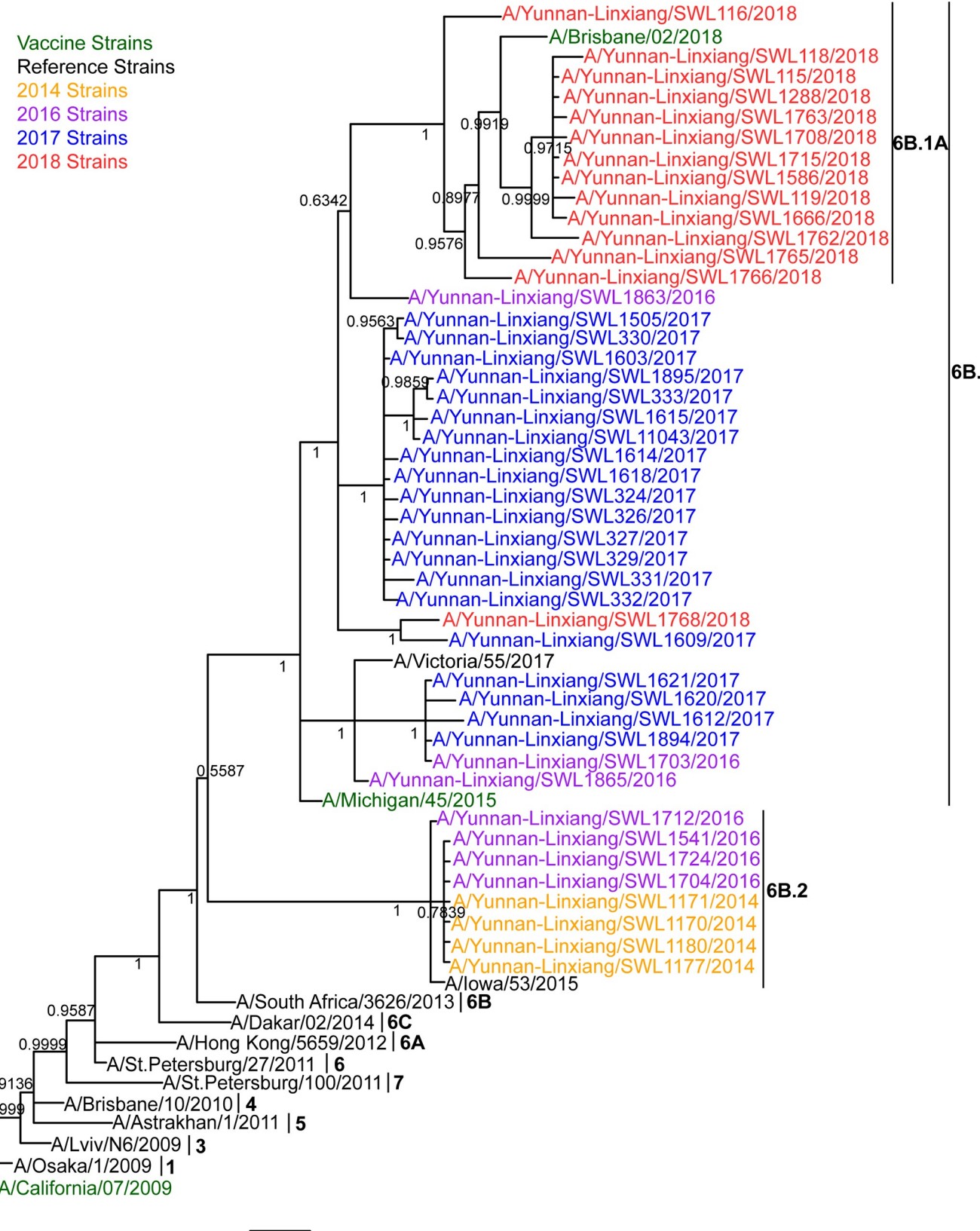

**Fig 2. Mrbayes phylogeny of the HA genes of influenza A(H1N1)pdm09 strains in Lincang, China, from 2014 to 2018.** 1, 3–7 and 6A~6B.1A indicated branch Numbers of clades. Each nodal number in phylogeny exhibited a Bayesian posterior probability (BPP). The ruler value (0.005) represented genetic distance.

RBS. Furthermore, other two substitutions (S183P and I216T) were also observed near RBSs that can increase receptor binding affinity [26–28]. These mutations found at antigenic sites and RBSs were further visualized to illustrate their potential functional effectiveness for the HA protein (Fig 3A and 3B).

Additionally, a MCCTREE under a MCMC uncorrelated relaxed molecular clock model using HA gene segments further supported the exact topology of Mrbayes phylogenetic tree and also provided a deep insight on the divergence time for ancestral nodes of the different clades (Fig 4). We discovered that the most recent common ancestors (TMRCAs) of clade 6B.1A, 6B.2 and all Lincang strains aroused in 2016.5683 year with 95% HPD [2016.1408, 2017.1150], 2013.5371 year with 95% HPD [2013.2442, 2013.9384] and 2012.4240 year with 95% HPD [2011.7507, 2013.2182], respectively. Meanwhile, based on BEAST v1.10.4, we estimated the mutation rates of all strain branches in Lincang with a median of $2.6x10^{-3}$ substitutions/site/year, and concluded that 2018 strains possessed a higher evolutionary rate (median: $2.95x10^{-3}$ substitutions/site/year) than that of other years (median: $2.40x10^{-3}$ substitutions/site/year) (S5 Table). Interestingly, we found that 2018 strains also accumulated the mutational amino acids more quickly than other years' strains (S2A Fig, Wilcoxon rank sum test, W = 123.5, P = 0.01285). And mutational amino acid statistics of each epitope of 2018 strains revealed that the ratio of amino acid substitutions in B epitope was much higher than those of other epitopes (C-E) (S2B Fig, Wilcoxon rank sum test, P<2.855e-06). These accumulated genomic variations in 2018 strains might potentially become dominant in the future.

## Phylogeny and mutation analysis of NA of influenza A(H1N1)pdm09 strains

The topology of the Mrbayes phylogenetic tree of NA genes from 45 strains was similar to that of HA (Fig 5 and S3 Fig). All the 45 strains clustered into 6B clade including representative substitutions (e.g. L40I, N44S, N200S, V241I, N248D, V264I, N270K, I321V, Y351F, N369K, N386K and K432E) (S6 Table). The strains of the 6B.2 lineage contained the unique substitutions V67I and T381I and the clade 6B.1 strains were defined by V13I, I34V and I314M. Moreover, most (15/20) strains from 2017 harbored one unique mutation E47G and the sub-clade 6B.1A shared the additional substitutions G77R (12/13)/G77K (1/13), V81A and N449D. And seven strains from clade 6B.1A possessed the extra substitutions T289I and D416G (S6 Table).

All the 45 strains carried eight conserved potential N-glycosylation sites in NA (positions 42, 50, 58, 63, 68, 88, 146 and 235), except that A/Yunnan-Linxiang/SWL1766/2018 harbored an additional substitution I23T, resulting in the acquisition of an N-glycosylation motif (NLT) (S6 Table). Like the Guangdong-A(H1N1)pdm09 viruses, the mutations V241I, N369K, N386K and K432E were also observed in Lincang strains which were reported to have significant impacts on the binding affinity between oseltamivir and neuraminidase, and further alter the susceptibility of A(H1N1)pdm09 strains to oseltamivir [29, 30]. Furthermore, the Lincang strains harbored the V106I (2/45) in 2017 and N248D (45/45) substitutions in NA, which could lead to low-PH stability of A(H1N1)pdm09 viruses and may contribute to its adaptation to human's immune system [9]. These mutations which were likely to bring about structural modification of NA protein of A(H1N1)pdm09 strains were plotted using PyMol (Fig 6).

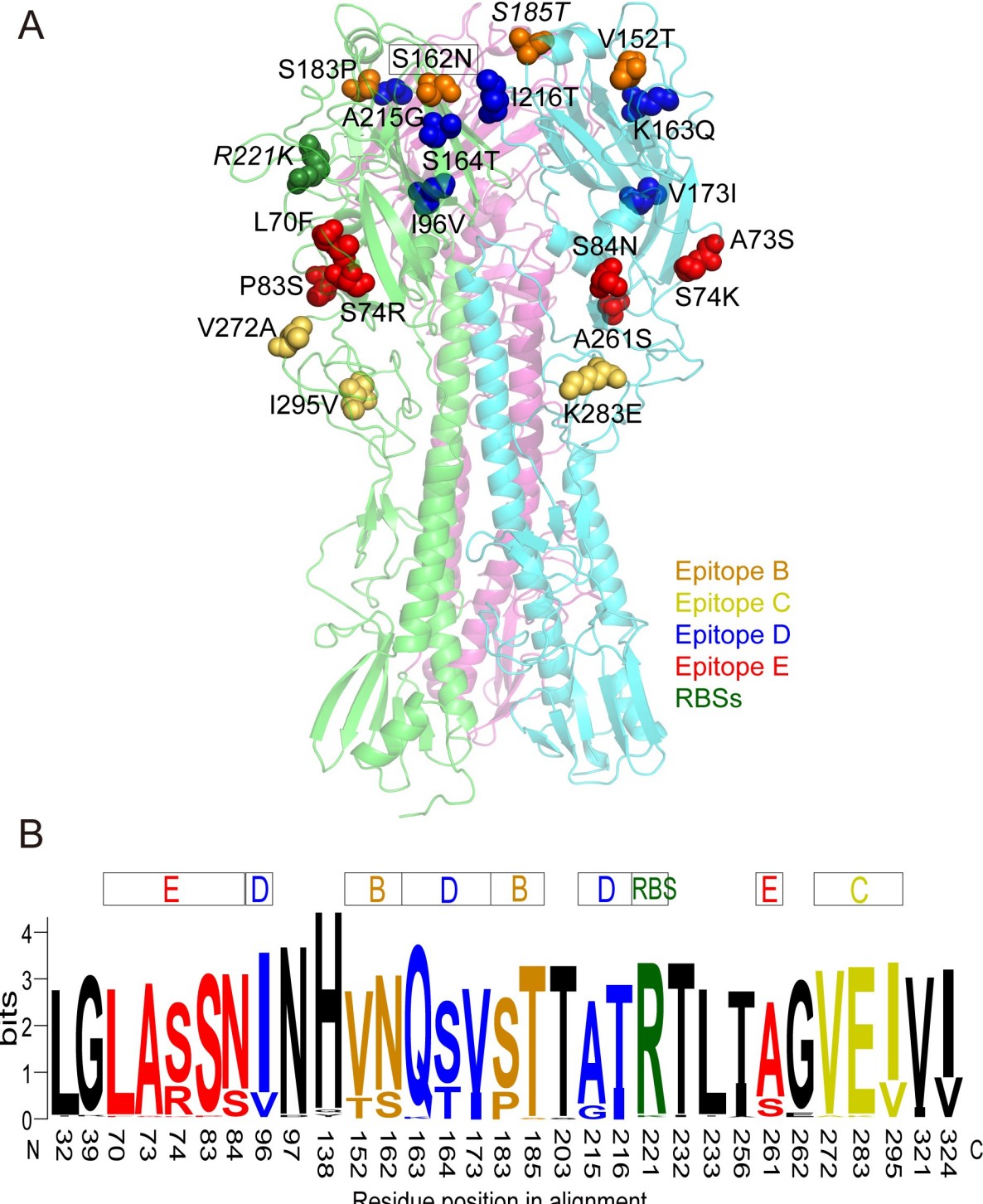

**Fig 3. Observed mutations and mutational frequencies of amino acid residues at the antigenic epitopes and RBSs of HA protein of influenza A(H1N1)pdm09 strains in Lincang, China, from 2014 to 2018.** (A) Observed mutations at the epitopes and RBSs of HA protein of influenza A(H1N1)pdm09 strains in Lincang, China, from 2014 to 2018. The amino acid differences at the epitopes and RBSs of HA, between Lincang 2014–2018 strains and vaccine strain (A/California/07/2009) were compared. We searched and obtained the model template (PDB ID: 6n41.1.A) of HA protein of A/California/07/2009. We conducted a structure prediction of the trimeric HA protein by SWISS-MODEL, then the changes at the epitopes and RBSs were visualized in PyMol. Amino acid substitutions located in different epitopes and RBSs were shown in different colors, respectively. The amino acid substitutions in RBSs were shown in italics, as well. Amino acid substitutions that were associated with the acquisition of a potential N-glycosylation site were framed. (B) Mutational frequencies of amino

acid residues at the antigenic epitopes and RBSs in the HA1 protein of A(H1N1)pdm09 strains in Lincang, China, from 2014 to 2018. The vaccine strain A/California/07/2009 was regarded as a reference. Relative frequency of the amino acid residues was proportional to the residue height. Mutational frequencies of amino acid changes at non-epitopes were shown in black color.

### Phylogeny and mutation analysis of six internal segments of influenza A (H1N1)pdm09 strains

Here, we also focused on genetic changes in the internal gene segments (i.e., M, NP, NS, PA, PB1 and PB2). Extremely similar to the Mrbayes phylogenetic trees of HA and NA, their tree topologies exhibited the similar evolutionary relationship (S4–S9 Figs).

The amino acid variations in internal proteins (M, NP, NS, PA, PB1 and PB2) also were analyzed (S7 Table). All the 45 strains were resistant to M2 inhibitors, owing to carrying the

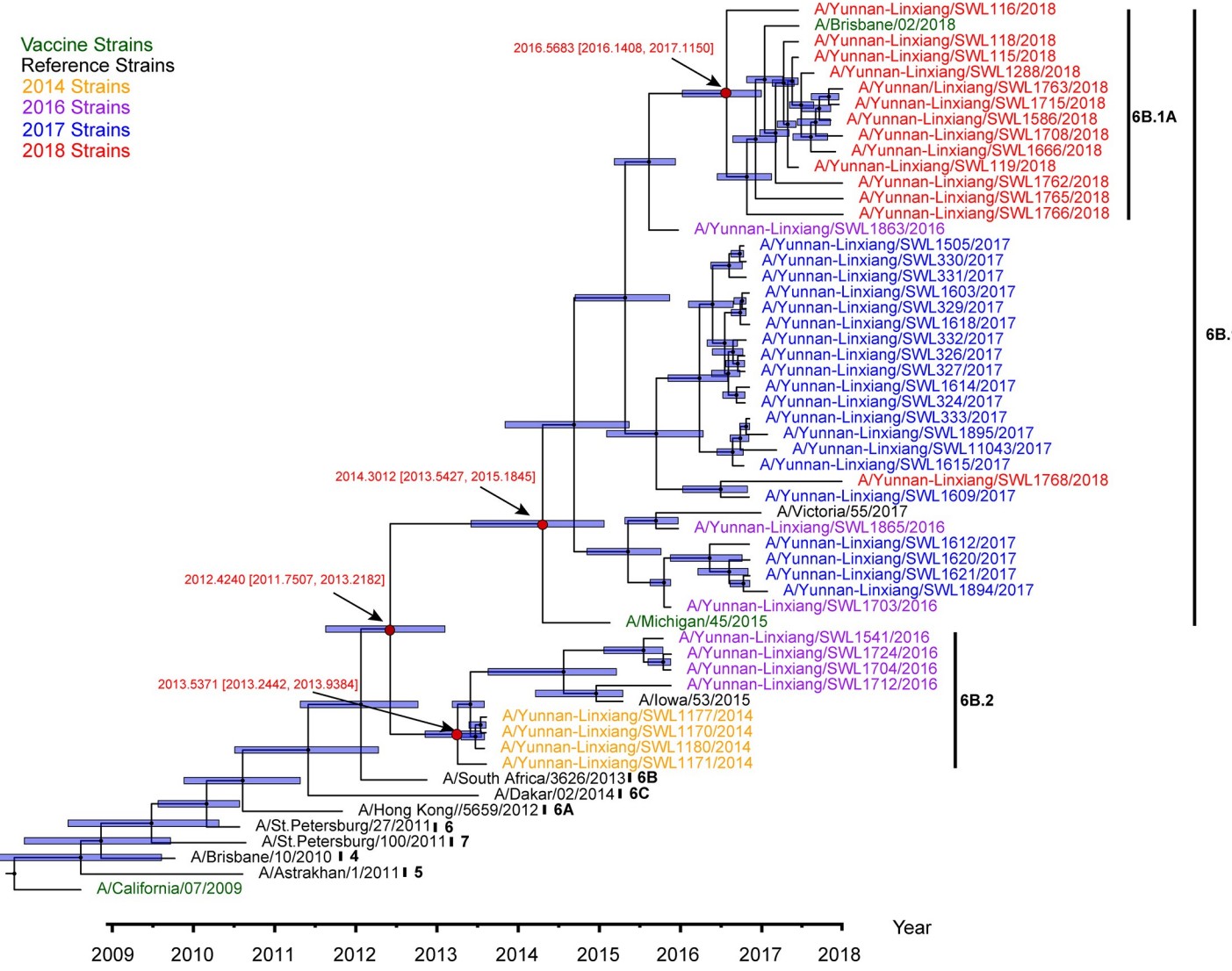

**Fig 4. Maximum clade credibility tree from HA alignment segments of influenza A(H1N1)pdm09 strains in Lincang, China, from 2014 to 2018.** The branch length represented time scale. The blue bars showed 95% HPD of nodal divergence time. TMRCAs of several nodes (clade 6B.1A, 6B.1, 6B.2 and all of isolated strains) were highlighted by the red font with 95% highest density probability (HPD). The two strains(A/Lviv/N6/2009 and A/Osaka/1/2009) without the sample collection date were removed in this analysis.

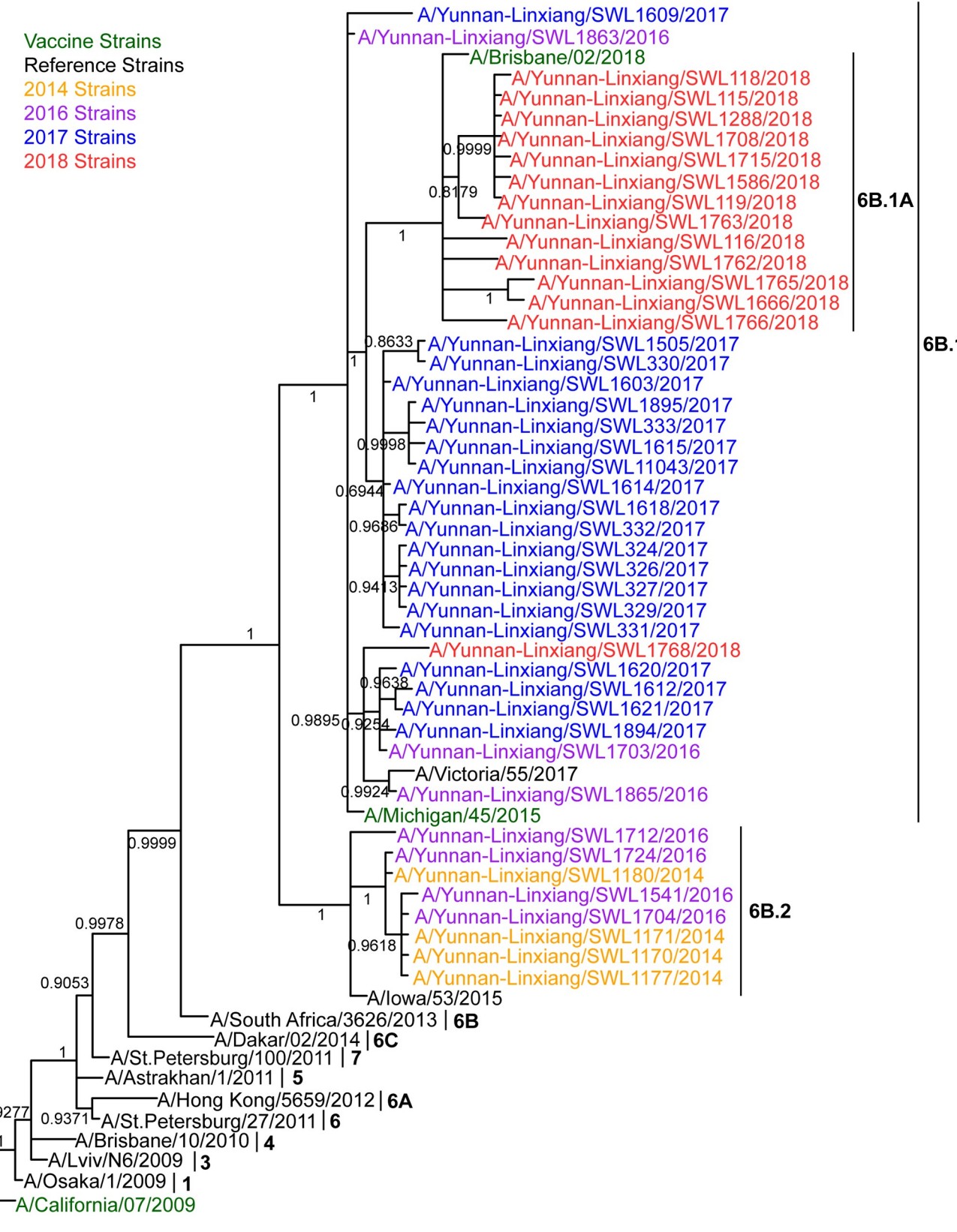

**Fig 5. Mrbayes phylogeny of the NA genes of influenza A(H1N1)pdm09 strains in Lincang, China, from 2014 to 2018.** 1, 3–7 and 6A~6B.1A indicated branch Numbers of clades. Each nodal number in phylogeny exhibited a Bayesian posterior probability (BPP). The ruler value (0.005) represented genetic distance.

mutation S31N in M2 [31] (S7 Table). Since 2009 pandemic, several mutations have been fixed in influenza A(H1N1)pdm09 viruses including M1-V80I, M1-M192V, M1-K230R, NP-A22T, NP-V100I, NP-M105T, NP-S498N, NS1-E55K, NS1-L90I, NS1-I123V, NS1-K131E, NS1-N205S, PA-V100I, PA-P224S, PA-N321K, PA-I330V, PA-R362K, PB1-G154D, PB1-I397M, PB1-I435T, PB2-R54K, PB2-M66I, PB2-D195N, PB2-R293K, PB2-V344M, PB2-I354L and PB2-V731I (S7 Table). In recent years (i.e., 2016–2018), newly accumulated mutations were observed in M1-Q208K, NS1-D2E, NS1-E125D, PB2-R299K and PB2-S453T, which were also the representative mutation sites of 6B.1 clades of their own phylogenetic trees. Additionally, two unique variations (PB1-R211K and PB1-K486R) were only identified in 2018 strains (9/13) (S7 Table).

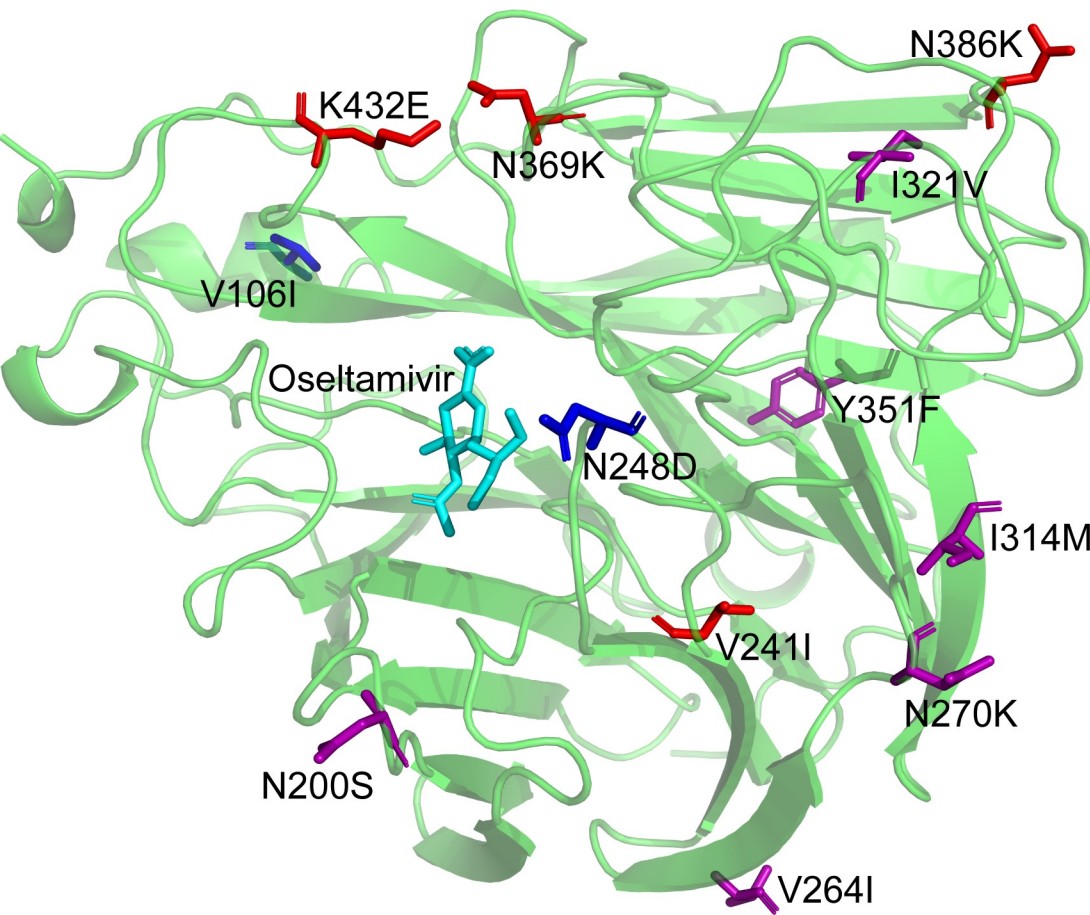

**Fig 6. Observed mutations in the important amino acid sites of NA protein of influenza A(H1N1)pdm09 strains in Lincang, China, from 2014 to 2018.** The amino acid differences in these important amino acid sites of NA between Lincang 2014–2018 strains and the vaccine strain (A/California/07/2009) were compared. The model template (PDB ID: 5nwe.1.A) of NA protein of A/California/07/2009 was searched and the tetramer structure of NA protein was predicted by SWISS-MODEL. Amino acid substitutions located in different amino acid sites were shown in different colors, respectively. Probable binding position of oseltamivir in NA was shown in light blue color. The common and conserved mutations were indicated in purple and the variations responsible for oseltamivir binding were shown in red, and the substitutions that were associated with low-PH stability of NA were displayed in blue color.

### Estimation of vaccine efficacy of influenza A(H1N1)pdm09 strains

Amino acid residues in five epitope regions A to E, which respectively possess 24, 22, 33, 48, and 34 amino acids, were defined by a previous report [20]. Table 1 provided a basic assessment of the predicted vaccine efficacy of vaccine strain against Lincang 2014–2018 strains.

With regards to vaccine strain A/California/07/2009, for 2014 and 2016, the $P_{epitope}$ between Lincang strains and vaccine strain A/California/07/2009 were 0.042–0.091 (dominant epitopes B, D and E) and 0.021–0.045 (dominant epitopes B-E), suggesting a predicated vaccine efficacy of 42.17%-48.00% and 47.65%-50.50%, respectively. For 2017 and 2018, the amino acid substitutions were observed both on epitopes B, D and E, resulting in the decreased vaccine efficacy of 42.17%-48.00%. However, when compared to the new vaccine strain A/Michigan/45/2015 recommended by WHO for 2017–2018, the $P_{epitope}$ declined to 0.021 and 0.021–0.03 (dominant epitopes D and C-E), yielding the increased vaccine efficacy ranging between 50.50% and 49.43%-50.50%. Notably, compared with the 2019 vaccine strain A/Brisbane/02/2018, the predicated vaccine efficacy against Lincang 2018 strains ranged of 45.74%-50.50%, with a slight descent. This suggests that the 2019 vaccine strain A/Brisbane/02/2018 may not fully provide modified protection against the circulating strains of 2018, in Lincang. It was mainly attributed to the additional substitutions (G45R, V298I and R223Q) on dominant epitopes C and D.

## Discussion

Consistent with global trends [7, 24, 32], our analyses from phylogenies using diverse gene segments indicate that all A(H1N1)pdm09 strains in Lincang, China, from 2014 to 2018, originated from the clade 6B, and deeply time-scale analyses reveal that the most recent common ancestor (TMRCA) of all Lincang strains in this study aroused in ~2012.4240 year. And recently the subclade 6B.1A has been predominated with a TMRCA in ~2016.5683 year for the local areas of southwest china.

At the molecular level, based on MCMC uncorrelated relaxed molecular clock model, for the first time, we produce an exact evaluation of mutation rate (median: $2.6 \times 10^{-3}$ substitutions/site/year) for all A(H1N1)pdm09 strains in Lincang, from 2014 to 2018. This evolutionary rate is obviously higher than that ($1.5$ or $1.6 \times 10^{-3}$) of pandemic (H1N1) 2009 influenza viruses in Sendai, Japan, during 2009–2011 [8]. However, it is significantly lower than that ($5.046 \times 10^{-3}$) of Saudi Arabia during the 2009–2011 pandemic seasons [33]. These imply that the mutation rate may potentially be influenced by regional diversification. Interestingly, we discover that Lincang strains in 2018 have evolved more rapidly (mutation rate: ~$2.95 \times 10^{-3}$ substitutions/site/year) than those strains in other years (mutation rate: ~$2.40 \times 10^{-3}$ substitutions/site/year). After integrating phylogenetic and antigenic epitope analyses, we indeed discover several important recently produced mutations especially in 2018 for diverse gene segments. For example, the mutation R221K (in 2018) is located at the RBS of HA, and it's well-known that previous studies have shown that mutations in or near the RBS domain of HA have important effects on the antigenic properties of influenza A(H1N1)pdm09 viruses [34, 35]. And another newly produced mutation S183P in HA covering 11/14 strains in 2018 and located at the epitope B is also near RBS, and proves to elevate receptor binding affinity [26–28]. For NA, A/Yunnan-Linxiang/SWL1766/2018 carries a new substitution I23T, leading to a potential gain of N-linked glycosylation site, and further functional experiments could highlight its importance in virus pathogenicity. Additionally, The NA segments of two Lincang strains in 2017 harbor a V106I mutation, and a previous study has indicated that this mutation has contributions to enhance the virulence of A(H1N1)pdm09 viruses in mice [36]. For other segments, two unique newly produced variations (i.e., PB1-R211K and PB1-K486R) were also harbored

**Table 1. Predicted vaccine efficacy of vaccine strains against influenza A(H1N1)pdm09 strains in Lincang, China, from 2014 to 2018.**

| Year(N) | Vaccine Strain | No. of strains | Dominant Epitope | No. of mutations | Residue differences | $P_{epitope}$ | Vaccine Efficacy(%) |
|---|---|---|---|---|---|---|---|
| 2014(N = 4) | A/California/07/2009 | **4** | **B** | **2** | **V152T,S185T** | **0.091** | **42.17** |
| | | 4 | C | 1 | K283E | 0.030 | 49.43 |
| | | **4** | **D** | **2** | **K163Q,V173I** | **0.042** | **48.00** |
| | | **4** | **E** | **2** | **P83S,A261S** | **0.059** | **45.98** |
| 2016(N = 7) | A/California/07/2009 | **7** | **B** | **1** | **S185T** | **0.045** | **47.65** |
| | | 4 | B | 1 | V152T | 0.045 | 47.65 |
| | | 3 | B | 1 | S162N | 0.045 | 47.65 |
| | | **7** | **C** | **1** | **K283E** | **0.030** | **49.43** |
| | | **7** | **D** | **1** | **K163Q** | **0.021** | **50.50** |
| | | 4 | D | 1 | V173I | 0.021 | 50.50 |
| | | 3 | D | 1 | I216T | 0.021 | 50.50 |
| | | 2 | D | 1 | A215G | 0.021 | 50.50 |
| | | **7** | **E** | **1** | **P83S** | **0.029** | **49.55** |
| | | 4 | E | 1 | A261S | 0.029 | 49.55 |
| | | 3 | E | 1 | S84N | 0.029 | 49.55 |
| 2017(N = 20) | A/California/07/2009 | **20** | **B** | **2** | **S162N,S185T** | **0.091** | **42.17** |
| | | 20 | C | 1 | K283E | 0.030 | 49.43 |
| | | **20** | **D** | **2** | **K163Q,I216T** | **0.042** | **48.00** |
| | | 4 | D | 1 | A215G | 0.021 | 50.50 |
| | | **20** | **E** | **2** | **P83S,S84N** | **0.059** | **45.98** |
| | | 1 | E | 1 | A73S | 0.029 | 49.55 |
| | A/Michigan/45/2015 | **4** | **D** | **1** | **A215G** | **0.021** | **50.50** |
| | | 1 | E | 1 | A73S | 0.029 | 49.55 |
| 2018(N = 14) | A/California/07/2009 | **14** | **B** | **2** | **S162N,S185T** | **0.091** | **42.17** |
| | | 11 | B | 1 | S183P | 0.045 | 47.65 |
| | | 14 | C | 1 | K283E | 0.030 | 49.43 |
| | | 13 | C | 1 | I295V | 0.030 | 49.43 |
| | | 1 | C | 1 | V272A | 0.030 | 49.43 |
| | | **14** | **D** | **2** | **K163Q,I216T** | **0.042** | **48.00** |
| | | 13 | D | 1 | S164T | 0.021 | 50.50 |
| | | 9 | D | 1 | I96V | 0.021 | 50.50 |
| | | **14** | **E** | **2** | **P83S,S84N** | **0.059** | **45.98** |
| | | 13 | E | 1 | S74R/K | 0.029 | 49.55 |
| | | 1 | E | 1 | L70F | 0.029 | 49.55 |
| | A/Michigan/45/2015 | 11 | B | 1 | S183P | 0.045 | 47.65 |
| | | **13** | **C** | **1** | **I295V** | **0.030** | **49.43** |
| | | 1 | C | 1 | V272A | 0.030 | 49.43 |
| | | **13** | **D** | **1** | **S164T** | **0.021** | **50.50** |
| | | 9 | D | 1 | I96V | 0.021 | 50.50 |
| | | **13** | **E** | **1** | **S74R/K** | **0.029** | **49.55** |
| | | 1 | E | 1 | L70F | 0.029 | 49.55 |
| | A/Brisbane/02/2018 | 3 | B | 1 | P183S | 0.045 | 47.65 |
| | | **14** | **C** | **2** | **G45R,V298I** | **0.061** | **45.74** |
| | | 1 | C | 1 | V272A | 0.030 | 49.43 |
| | | 1 | C | 1 | V295I | 0.030 | 49.43 |
| | | **14** | **D** | **1** | **R223Q** | **0.021** | **50.50** |
| | | 9 | D | 1 | I96V | 0.021 | 50.50 |
| | | 1 | D | 1 | T164S | 0.021 | 50.50 |
| | | 1 | E | 1 | L70F | 0.029 | 49.55 |
| | | 2 | E | 1 | R74K/S | 0.029 | 49.55 |

Dominant epitopes are shown in Bold.

by 2018 strains with a high coverage (9/13), suggesting potential functional candidates in the future.

The rapid mutation rate of influenza A(H1N1)pdm09 in Lincang especially in recent years poses an enormous challenge for the timely renewal of vaccines. Our analyses in vaccine efficacy consolidate the WHO decision to update northern hemisphere A(H1N1)pdm09 vaccine strain in 2017 [37–39]. As expected, the estimated vaccine efficacy between A/Michigan/45/2015 vaccine strain and the 2017 circulating strains in Lincang increased to 50.50%. Subsequently, the estimated vaccine efficacy against Lincang 2018 strains dropped to a range of 49.43%~50.50%, which further suggests a rapid evolution of strains in 2018. Although the estimated vaccine efficacy of A/Brisbane/02/2018 vaccine strain has not improved as expected, our analyses do not include the isolated strains in 2019 and cannot directly reflect a true vaccine efficacy against the circulating strains in 2019.

## Conclusions

In conclusion, consistent phylogenetic trees and those functional mutations elucidate the molecular mechanisms of origin and rapid evolution of the influenza A(H1N1)pdm09 strains in Lincang, China, from 2014 to 2018. And we also verifies the need for constant molecular monitoring of the circulating strains to provide prompt supporting data for the better selection of the vaccine strain of influenza A(H1N1)pdm09.

## Supporting information

**S1 Table. Accession numbers in GISAID databases of applied gene segments of 14 A (H1N1)pdm09 reference strains included in the analysis.** N/A: not applied.
(DOC)

**S2 Table. The number (%) of influenza strains isolated from samples of influenza-like cases in Lincang, China, from 2014 to 2018.**
(DOC)

**S3 Table. Demographic characteristics of the cases infected with influenza viruses in Lincang, China, from 2014 to 2018.**
(DOC)

**S4 Table. Amino acid substitutions in the epitopes and non-epitopes of HA protein of influenza A(H1N1)pdm09 strains in Lincang, China, from 2014 to 2018.** Amino acid mutations were referred to A/California/07/2009. "." showed that the amino acid site was the same as that of A/California/07/2009. "a" represented the receptor binding sites. "+" indicated the gain of a potential glycosylation site. "-" indicated the predicated deleterious mutations.
(XLSX)

**S5 Table. Estimated evolutionary rates of HA genes of influenza A(H1N1)pdm09 strains in Lincang, China, from 2014 to 2018.** The evolutionary rates were estimated by BEAST v1.10.4 software under the uncorrelated relaxed clock model.
(DOC)

**S6 Table. Amino acid substitutions of NA protein of influenza A(H1N1)pdm09 strains in Lincang, China, from 2014 to 2018.** Amino acid mutations were referred to A/California/07/2009. "." showed that the amino acid site was the same as that of A/California/07/2009. "+" indicated the gain of a potential glycosylation site. "-"indicated the predicated deleterious mutations.
(XLSX)

**S7 Table. Important amino acid substitutions of six internal proteins of influenza A (H1N1)pdm09 strains in Lincang, China, from 2014 to 2018.** Amino acid mutations were referred to A/California/07/2009. "." showed that the amino acid site was the same as that of A/California/07/2009.
(XLSX)

**S1 Fig. Mrbayes phylogeny of the HA genes of influenza A(H1N1)pdm09 strains in Lincang, China, from 2014 to 2018.** Each nodal number in square brackets represented a Bayesian posterior probability (BPP) range. The ruler value (0.005) represented genetic distance.
(TIF)

**S2 Fig. Comparative analyses of mutations in HA protein between 2018 and other years.** (A) Comparative analyses of accumulated variations in HA between 2018 and other year strains. (B) Comparative analyses of mutations under different epitopes for HA in 2018 strains. The significant level was decided by Wilcoxon rank sum test (P<0.05).
(TIF)

**S3 Fig. Mrbayes phylogeny of the NA genes of influenza A(H1N1)pdm09 strains in Lincang, China, from 2014 to 2018.** Each nodal number in square brackets represented a Bayesian posterior probability (BPP) range. The ruler value (0.005) represented genetic distance.
(TIF)

**S4 Fig. Mrbayes phylogeny of the M genes of influenza A(H1N1)pdm09 strains in Lincang, China, from 2014 to 2018.** 6B~6B.1A indicated branch Numbers of clades. Each nodal number in phylogeny exhibited a Bayesian posterior probability (BPP). The ruler value (0.2) represented genetic distance.
(TIF)

**S5 Fig. Mrbayes phylogeny of the NP genes of influenza A(H1N1)pdm09 strains in Lincang, China, from 2014 to 2018.** 6B~6B.1A indicated branch Numbers of clades. Each nodal number in phylogeny exhibited a Bayesian posterior probability (BPP). The ruler value (0.002) represented genetic distance.
(TIF)

**S6 Fig. Mrbayes phylogeny of the NS genes of influenza A(H1N1)pdm09 strains in Lincang, China, from 2014 to 2018.** 6B~6B.1A indicated branch Numbers of clades. Each nodal number in phylogeny exhibited a Bayesian posterior probability (BPP). The ruler value (0.2) represented genetic distance.
(TIF)

**S7 Fig. Mrbayes phylogeny of the PA genes of influenza A(H1N1)pdm09 strains in Lincang, China, from 2014 to 2018.** 6B~6B.1A indicated branch Numbers of clades. Each nodal number in phylogeny exhibited a Bayesian posterior probability (BPP). The ruler value (0.002) represented genetic distance.
(TIF)

**S8 Fig. Mrbayes phylogeny of the PB1 genes of influenza A(H1N1)pdm09 strains in Lincang, China, from 2014 to 2018.** 6B~6B.1A indicated branch Numbers of clades. Each nodal number in phylogeny exhibited a Bayesian posterior probability (BPP). The ruler value (0.002) represented genetic distance.
(TIF)

**S9 Fig. Mrbayes phylogeny of the PB2 genes of influenza A(H1N1)pdm09 strains in Lincang, China, from 2014 to 2018.** 6B~6B.1A indicated branch Numbers of clades. Each nodal number in phylogeny exhibited a Bayesian posterior probability (BPP). The ruler value (0.002) represented genetic distance.
(TIF)

## Acknowledgments

The authors thank the staff of the hospital and CDC in Lincang for their assistance with the collection of specimens of influenza-like illness and virus isolation and identification.

## Author Contributions

**Conceptualization:** Mei-Ling Zhang.

**Formal analysis:** Xiao-Nan Zhao, Yong Shao.

**Methodology:** Yong Shao.

**Software:** Yong Shao.

**Validation:** Han-Ju Zhang, Duo Li, Jie-Nan Zhou, Yao-Yao Chen, Yan-Hong Sun.

**Visualization:** Yong Shao.

**Writing – review & editing:** Adeniyi C. Adeola, Xiao-Qing Fu, Yong Shao, Mei-Ling Zhang.

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
