## [Decision Letter · Decision Letter 0]

18 Feb 2020

PONE-D-20-01217

Whole-Genome Sequencing Reveals Origin and Evolution of Influenza A(H1N1)pdm09 Viruses in Lincang, China, from 2014 to 2018

PLOS ONE

Dear Mrs Zhang,

Thank you for submitting your manuscript to PLOS ONE. After careful consideration, we feel that it has merit but does not fully meet PLOS ONE’s publication criteria as it currently stands. Therefore, we invite you to submit a revised version of the manuscript that addresses the points raised during the review process.

I agree with the experts who have evaluated the manuscript that the content presented  in the manuscript is timely and very relevant to current influenza infection. The expert reviewers proposed several modifications. I believe addressing these comments will greatly enhance the scientific quality  and will improve the readability of the manuscript. Therefore, please address both reviewer's comments carefully and incorporate their suggestions to make the manuscript suitable for publication.

We would appreciate receiving your revised manuscript by Apr 03 2020 11:59PM. To enhance the reproducibility of your results, we recommend that if applicable you deposit your laboratory protocols in protocols.io, where a protocol can be assigned its own identifier (DOI) such that it can be cited independently in the future. For instructions see: http://journals.plos.org/plosone/s/submission-guidelines#loc-laboratory-protocols

We look forward to receiving your revised manuscript.

Kind regards,

Mrinmoy Sanyal, PhD

Academic Editor

PLOS ONE

Journal Requirements:

1. Please provide additional details regarding participant consent. In the ethics statement in the Methods and online submission information, please ensure that you have specified whether consent was written or verbal/oral. If consent was verbal/oral, please specify:

a) whether the ethics committee approved the verbal/oral consent procedure,

b) why written consent could not be obtained, and

c) how verbal/oral consent was recorded. If your study included minors, please state whether you obtained consent from parents or guardians in these cases.

https://doi.org/10.1002/jmv.25328

In your revision ensure you cite all your sources (including your own works), and quote or rephrase any duplicated text outside the Methods section. Further consideration is dependent on these concerns being addressed.

Reviewers' comments:

Reviewer's Responses to Questions

**Comments to the Author**

1. Is the manuscript technically sound, and do the data support the conclusions?

Reviewer #1: Yes

Reviewer #2: Yes

2. Has the statistical analysis been performed appropriately and rigorously? 

Reviewer #1: Yes

Reviewer #2: N/A

3. Have the authors made all data underlying the findings in their manuscript fully available?

Reviewer #1: Yes

Reviewer #2: Yes

4. Is the manuscript presented in an intelligible fashion and written in standard English?

Reviewer #1: No

Reviewer #2: Yes

5. Review Comments to the Author

Reviewer #1: This is an important paper, about evolution of H1N1 sequences from regions often thought to be the source of new influenza strains. I have a few comments

1) It would be helpful when pEpitope is introduced to mention the relation between Sa, Sb, Ca1, Ca2, and Cb and epitopes A-E. The mapping is "Sa, Sb, Ca1 , Ca2 and Cb" -> "B+D, B, C+D, A+D and E". Plus the A-E epitopes contain an additional 31 amino acids.

2) Since pEpitope is used to compute vaccine efficacy, it would be more helpful to identify epitope membership rather than the Sa, Sb, Ca1, Ca2, and Cb membership in Table 1 and Figure 4. Or, this could be done in SI figures.

3) Please verify the numbering from Table 1 of ref 21 was used to compute pEpitope (or that the .xls or matlab script was used to compute pEpitope).

3) How are years defined? Usually, we group influenza strains per season (e.g. from July 1 year n to June 30 year n+1). I am not sure whether seasonal grouping is correct for China, however, as new strains seem to arise in China in the summer.

4) Regarding this sentence: "Our results suggested that a timely replacement of the vaccine was absolutely necessary because that the mean vaccine efficacy against the circulating strains in 2018 showed a decreased trend, compared to 2017" it would be helpful to compare the 2017 and 2018 strains against the old vaccine (California) to show the new vaccine (Michigan) was expected to have higher efficacy. It would also be interesting to compare the 2018 strains against the newer vaccine (Brisbane) to show Brisbane would be expected to have higher efficacy.

5) It would be interesting to describe in which epitope most of the evolution in 2018 (e.g. figure S1) ocurred.

Reviewer #2: The manuscript by Zhang et al, has described the genomic sequencing of influenza A(pH1N1) viruses from Lincang in China from 2014 to 2018. Samples for this study are sourced from ILI surveillance and WGS was conducted from cell culture isolated virus. While this work and findings are important and relevant, the key messages is lost in the verboseness of the manuscript.

Methods:

1. The methods used appear to be standard methods for China, The authros must refence literature to provide a description of methods – realtime RT-PCR, virus isolation, HA and HI. What was the source of the strain identification kit?

2. Authors mention genome amplification using “Invitrogen SuperScript III One-Step RT-PCR Platinum kit”. IT is not clear what is amplified. The primers used must be stated,

3. What was the protocol for library prep? Should be mentioned briefly.

4. While the authros have states use of a MiSeq for generation of whole genome sequence. A description of sequencing kit used and the run conditions and protocol for data handling is not mentioned.

5. Any reason why contigs were generated de novo? The usual protocol for influenza is to map to reference strains

6. Details on the “Global Reference Strains” are missing.

7. For the phylogenetic analysis, what was the nucleotide substitution model used? This has a bearing on the results

8. What is the tree being presented? MCC tree? Software used for annotation of trees?

9. Vaccine efficacy (16,22,23)

Results

1. This study is a detailed sequencing analysis of influenza A /pH1N1 strains obtained from routine ILI surveillance. However, figure 1 only has details fo the number of strains and seuqneing strains. How many samples were collected and what was the detection rate of influenza ( A and B) in this study? Demographic profile of influenza is not mentioned.

2. The source of these strains as mentioned by the authors is not clear.

3. Authors infer “faster evolutionary rate than other years (Supplementary Fig S1, Wilcoxon rank sum test, W=123.5, P=0.01285)”. However, the parameters being compared are ratio of mutational sites/total sites.

4. The HA and NA phylogeny with listed mutation sites is verbose and difficult to read. It will benefit from a table summarizing the mutations.

5. Has the tMRCA been calculated?

6. HPD of nodes must be presented.

7. Genetic variation presented in the phylogeny has again been presented in the context of the HA antigenic sites and NA sites conferring neuraminidase resistance.

8. The relevance of mutations belongs in the discussion but is presented in the results.

9. THe prediction of “vaccine efficacy” is confusing to read. This section needs to be reworked to allow the reader to understand what terms like “worst-case” efficacy and “perfect match”mean. The table is confusing and must either be removed or presented in a more intelligible fashion.

Discussion: The discussion does not highlight the relevance of the work. The discussion needs to be reworked to summarize key findings of the study and their agreement in a functional or epidemiological or evolutionary framework. Some of the content in the results belongs in the discussion e.g the paragraph on pg 19 “Several substitutions listed above…” etc.

6. PLOS authors have the option to publish the peer review history of their article (what does this mean?). If published, this will include your full peer review and any attached files.

Reviewer #1: No

Reviewer #2: No

---

## [Author Response · Author response to Decision Letter 0]

10 Apr 2020

April 1, 2020

To

The Editorial Office,

PloS One

Dear editors and reviewers:

Thank you for your kind reply. We are very grateful for the reviewer’s comments on our manuscript entitled “Whole-genome sequencing reveals origin and evolution of influenza A(H1N1)pdm09 viruses in Lincang, China, from 2014 to 2018” (PONE-D-20-01217). Indeed, your comments are very helpful in improving our manuscript. We have carefully studied your comments point by point and tried our best to revise the manuscript to meet your approvals. We believe that we have settled all of the concerns from the reviewers, and the revisions are documented in the revised manuscript and supplementary information. The responses to the reviewers’ comments are listed as follows (Our replies are in blue).

Kind regards,

Corresponding author: Mei-Ling Zhang

E-mail address: meilingz2011@163.com

Responses to reviewers' comments

Reviewer's Responses to Questions

Comments to the Author

1. Is the manuscript technically sound, and do the data support the conclusions?

Reviewer #1: Yes

Reviewer #2: Yes

2. Has the statistical analysis been performed appropriately and rigorously? 

Reviewer #1: Yes

Reviewer #2: N/A

Reply:

For statistical analysis, we have re-normlized by scientific algorithms and softwares. Please check our improvements.

3. Have the authors made all data underlying the findings in their manuscript fully available?

Reviewer #1: Yes

Reviewer #2: Yes

4. Is the manuscript presented in an intelligible fashion and written in standard English?

Reviewer #1: No

Reviewer #2: Yes

Reply:

We have done our best to reduce the specific errors and discuss with local English professors. Please check our revised manuscript.

5. Review Comments to the Author

Reply:

Thank you very much. According to your requests, we have provided consistent answers.

Reviewer #1: This is an important paper, about evolution of H1N1 sequences from regions often thought to be the source of new influenza strains. I have a few comments

1) It would be helpful when pEpitope is introduced to mention the relation between Sa, Sb, Ca1, Ca2, and Cb and epitopes A-E. The mapping is "Sa, Sb, Ca1 , Ca2 and Cb" -> "B+D, B, C+D, A+D and E". Plus the A-E epitopes contain an additional 31 amino acids.

Reply:

Thank you very much for the professional suggestions. And we fully agree with your points. Now, according to your suggestions, we have adjusted our revisions as follows:

(1) To make our manuscript more clearer, when we discussed the antigenic epitopes, we explained the theory of the mapping relationships from Sa, Sb, Ca1, Ca2 and Cb to B+D, B, C+D, A+D and E. According to the sequence homology, mutations observed in five epitopes (Sa, Sb, Ca1, Ca2, and Cb) were mapped to the H3 epitopes (B+D, B, C+D, A+D and E), respectively. So, we changed memberships (Sa, Sb, Ca1, Ca2 and Cb) into epitope memberships (B+D, B, C+D, A+D and E).

(2) And further we cited the previous published references to illustrate that these mapping relationships are reliable (Deem and Pan 2009; Opanda, et al. 2020).

(3) These mapping relationships provided a preliminary basis to understand the downstream evaluation of Pepitope.

Please check our revised Result section << Phylogeny and mutation analysis of HA of influenza A(H1N1)pdm09 strains>>. We hope that our revisions can meet your criterion.

2) Since pEpitope is used to compute vaccine efficacy, it would be more helpful to identify epitope membership rather than the Sa, Sb, Ca1, Ca2, and Cb membership in Table 1 and Figure 4. Or, this could be done in SI figures.

Reply:

We thank the reviewer for pointing out this issue. Consistent with above answers, we have identified the epitope relationship in our revised Result section << Phylogeny and mutation analysis of HA of influenza A(H1N1)pdm09 strains>> to improve our manuscript. In detail, the mutation sites and 3D structure analyses of HA protein have been adjusted by the mapped five epitopes (A-E), and please see our revised S4 Table and Fig 3.

3) Please verify the numbering from Table 1 of ref 21 was used to compute pEpitope (or that the .xls or matlab script was used to compute pEpitope).

Reply:

Thanks for your good comments. According to your points, we repeatly validated the numbering and our numbering (Table 1) in this study was accordance with your listed research, which point out that the five epitopes (A to E) respectively possess 24, 22, 33, 48, and 34 amino acids (Deem and Pan 2009). The residue differences on the epitopes were counted and reanalyzed in our Table 1. 

Further, the vaccine efficacy was estimated by the formula (E = 0.47–2.47 x Pepitope). Also please check our revised method, especially for this section << Estimation of vaccine efficacy of influenza A(H1N1)pdm09 strains>>. And three related references were cited in this section accordingly (Deem and Pan 2009; Gupta, et al. 2006; Monamele, et al. 2019).

3) How are years defined? Usually, we group influenza strains per season (e.g. from July 1 year n to June 30 year n+1). I am not sure whether seasonal grouping is correct for China, however, as new strains seem to arise in China in the summer.

Reply:

We thank the reviewer to point out this issue. We agree with your views that generally you group influenza strains from July 1 year n to June 30 year n+1. Herein, in this study we defined one year from January 1 to December 31. Thus this manuscript has covered the period from January 1, 2014 to December 31, 2018. 

Please see our revised sampling strategy in Results. We added according description in this revised manuscript.

4) Regarding this sentence: "Our results suggested that a timely replacement of the vaccine was absolutely necessary because that the mean vaccine efficacy against the circulating strains in 2018 showed a decreased trend, compared to 2017" it would be helpful to compare the 2017 and 2018 strains against the old vaccine (California) to show the new vaccine (Michigan) was expected to have higher efficacy. It would also be interesting to compare the 2018 strains against the newer vaccine (Brisbane) to show Brisbane would be expected to have higher efficacy.

Reply:

Thanks for your good suggestions to improve this article. We have compared the 2017 and 2018 strains against the old vaccine strain (A/California/07/2009). Judging from the results, there was an expected increase (from 24.52%-36.63% to 39.59%-41.81%) in vaccine efficacy when the vaccine strain changed to A/Michigan/45/2015 in 2017-2018. 

Further, we also compared the 2018 strains against the newer vaccine strain (A/Brisbane/02/2018), but the predicated vaccine efficacy declined to the ranging of 31.93%-41.81%. It was mainly attributed to the additional three substitutions (G45R, V298I and R223Q) on dominant epitopes C and D. We also observe that the vaccine efficacy against 2018 strains was lower than that of 2017. Therefore, it’s essential for the WHO to update the vaccine strain in 2019. However, due to the time lag of research, our study did not include the strains isolated in 2019, so it could not directly reflect the true vaccine efficacy against the circulating strains of 2019, in lincang, China. On the other hand, WHO determined the vaccine strain compositions based on the situation of global circulating strains, so the vaccine strains recommended by WHO could not always provide better vaccine efficacy in local city, which was also supported by previous studies (Opanda, et al. 2020). 

Please check our revised Results section <<Estimation of vaccine efficacy of influenza A(H1N1)pdm09 strains>> in detail.

5) It would be interesting to describe in which epitope most of the evolution in 2018 (e.g. figure S1) ocurred.

Reply:

Thanks for your valuable advices. According to your good suggestions, we have analyzed the amino acid replacement rate for each epitope of 2018 strains. Our results revealed that the epitope B accumulated mutational amino acids more quickly than that of other epitopes (C-E) (Wilcoxon rank sum test, P<2.855e-06). The data have been described in S2B Fig.

Reviewer #2: The manuscript by Zhang et al, has described the genomic sequencing of influenza A(pH1N1) viruses from Lincang in China from 2014 to 2018. Samples for this study are sourced from ILI surveillance and WGS was conducted from cell culture isolated virus. While this work and findings are important and relevant, the key messages is lost in the verboseness of the manuscript.

Reply:

We thank the reviewer for pointing out these issues. According to you suggestions, we have provided careful improvements in this revised manuscript. We hope that our revisions can meet your criteria well.

Methods:

1. The methods used appear to be standard methods for China, The authros must refence literature to provide a description of methods – realtime RT-PCR, virus isolation, HA and HI. What was the source of the strain identification kit?

Reply: 

Thanks for your valuable comments. According to your suggestions, we have cited Technical Guidelines for National Influenza Surveillance (2017 Edition) (http://www.chinaivdc.cn/cnic/zyzx/jcfa/201709/t20170930_153976.htm) to provide the detailed experimental protocols. The strain identification kit was furnished by Chinese National Influenza Center, and the post-infection ferret antisera was raised by corresponding vaccine strains (A/California/7/2009(H1N1)pdm09 for 2009-2016 and A/Michigan/45/2015 for 2017-2018) and selected representative strains in China. 

The related information has been supplemented in Materials and methods section << Specimen collection and isolation and identification of virus>>.

2. Authors mention genome amplification using “Invitrogen SuperScript III One-Step RT-PCR Platinum kit”. IT is not clear what is amplified. The primers used must be stated,

Reply: 

Thanks for your suggestions. According to your points, we have provided the information of genome amplification. For example, the SuperScriptTM III One-Step RT-PCR System with PlatinumTM Taq High Fidelity was used to perform cDNA synthesis and PCR amplification of viral nucleic acids. 

Next, the amplification primers included Uni-12/Inf1 (primer A): 5’-GGGGGGAGCAAAAGCAGG-3’, Uni-12/Inf3 (primer B): 5’-GGGGGGAGCGAAAGCAGG-3’ and Uni-13/Inf1 (primer C): 5’-CGGGTTATTAGTAGAAACAAGG-3’. They have been added in the revised Materials and methods section <<Whole genome sequencing of influenza A(H1N1)pdm09>>.

3. What was the protocol for library prep? Should be mentioned briefly.

Reply: 

Thanks very much for your kind points. According to your suggestions, we have added a clear description to our revised Materials and methods section <<Whole genome sequencing of influenza A(H1N1)pdm09> to briefly describe the protocols for library prep. Please check our revised manuscript.

4. While the authros have states use of a MiSeq for generation of whole genome sequence. A description of sequencing kit used and the run conditions and protocol for data handling is not mentioned.

Reply: 

Thanks a lot. We have provided improvements for the Materials and methods section <<Whole genome sequencing of influenza A(H1N1)pdm09> in this revised manuscript. To make our description more clearer, we described as follows in our revised manuscript:

“In the end, we thawed reagent cartridge (MiSeq v2 Reagent Tray 300 cycles – PE) , loaded the pooled libraries onto the reagent cartridge in the designated reservoir and operated MiSeq sequencer applied to sequencing according to the illumine MiSeq system guide. The raw sequencing paired-end reads were imported to CLC Genomics Workbench 9.5.2 (Qiagen, Denmark) and further were trimmed and performed for de novo assembly into contigs. ”

5. Any reason why contigs were generated de novo? The usual protocol for influenza is to map to reference strains

Reply: 

Thank you so much. We are in agreement with your comments. As a matter of fact, this has been done according to the usual protocol. We apologize for our unclear descriptions. Now, in this revised manuscript, we have added this part to our updated version. In detail, the raw sequencing paired-end reads were imported to CLC Genomics Workbench 9.5.2 (Qiagen, Denmark) and further were trimmed and performed for de novo assembly into contigs. Then, the contigs were imported to National Center for Biotechnology Information (https://www.ncbi.nlm.nih.gov/genomes/FLU/Database/nph-select.cgi?go=database) for blast, and the best-hit reference strains were selected for mapping to produce consensuses.

6. Details on the “Global Reference Strains” are missing.

Reply: 

Thanks very much. We have revised it in this manuscript. The 14 global reference strains included three northern hemisphere vaccine strains (2014-2020) recommended by WHO. And the detail information (accession numbers) about global reference strains was replenished in S1 Table.

7. For the phylogenetic analysis, what was the nucleotide substitution model used? This has a bearing on the results

Reply: 

Thanks. The phylogenetic trees were constructed using Mrbayes method under the HKY+I+G nucleotide substitution model. This has been added to the methods section in our revised manuscript.

Additionally, we also performed the MCCTREE analyses by BEAST v1.10.4 software to estimate divergence time and obtained an according topology (please see below answers for MCCTREE). This suggests our analyses including our model parameters are reliable. 

8. What is the tree being presented? MCC tree? Software used for annotation of trees?

Reply: 

Many thanks. We apologize for our unclear statements although we have described our Mrbayes algorithm applied to construct phylogenies in Materials and methods. Here, in the revised figure legends (Figs 2 and 5), we also added the ‘Mrbayes phylogeny’ to make our descriptions more clearly. Please check the according revision in main text.

The Mrbayes phylogenetic trees were presented and annotated in Fig Tree v1.4.3 software. This has also been appended to the method section in our revised manuscript.

9. Vaccine efficacy (16,22,23)

Reply: 

Thank you for your professional advice. We have re-calculated the vaccine efficacy based on a published reference (Deem and Pan 2009). The results were in good agreement with the trends obtained from your listed references (Falchi, et al. 2011; Pan, et al. 2011; Tewawong, et al. 2015).

Results

1. This study is a detailed sequencing analysis of influenza A /pH1N1 strains obtained from routine ILI surveillance. However, figure 1 only has details fo the number of strains and seuqneing strains. How many samples were collected and what was the detection rate of influenza ( A and B) in this study? Demographic profile of influenza is not mentioned.

Reply: 

Thanks greatly for the useful comment. We calculated and supplemented this part in Results section <<Isolation rate, demographic characteristic and sampling distribution of influenza>> in this revised manuscript. 6767 influenza-like illness samples were collected in Lincang people's hospital, during the study period (January 1, 2014 to December 31, 2018). The isolation rates of influenza A and B strains were 3.04% and 2.75%, respectively. Among the 392 cases infected with influenza viruses, male accounted for the majority (61.73%), mainly in the age group under 5 years old, and scattered children were the main infected population. The detailed data were shown in S2 Table.

2. The source of these strains as mentioned by the authors is not clear.

Reply: 

Thanks a lot. First, all the 6767 influenza-like illness samples were collected in Lincang people's hospital, which is the only sentinel surveillance hospitals in Lincang. Second, total 392 strains were isolated from the 6767 influenza-like illness samples. Among that, 112 strains were influenza A(H1N1)pdm09 strains. In the end, we randomly selected 45 strains for subsequent sequencing experiments. Please further check our revisions in Result section <<Isolation rate, demographic characteristic and sampling distribution of influenza>> in this revised manuscript.

3. Authors infer “faster evolutionary rate than other years (Supplementary Fig S1, Wilcoxon rank sum test, W=123.5, P=0.01285)”. However, the parameters being compared are ratio of mutational sites/total sites.

Reply:

We are grateful to the reviewer for normalizing our vague statements for this section. I’m sorry for unclear descriptions. Actually, our analyses in this section only illustrated that the strains in 2018 have accumulated amino acid variations more quickly than those from all of other years. Therefore, we rephrased our descriptions in this revised manuscript. Additionally, we further provided insights into the statistics for different antigenic epitopes about amino acid substitutions. Please check our revisions in main text accordingly.

Importantly, according to your suggestions, we also provided accurate estimates of virus evolutionary rates (substitutions/site/year) for each branch in our phylogeny using BEAST v1.10.4 under an uncorrelated relaxed molecular clock model. We supplied this information to S5 Table including the evolutionary rates from all of isolated strains. Please check our revisions.

4. The HA and NA phylogeny with listed mutation sites is verbose and difficult to read. It will benefit from a table summarizing the mutations.

Reply: 

Thanks. According to your suggestions, we have deleted the listed mutation sites in HA and NA phylogeny, as can be seen in Figs 2 and 5. The mutations of HA and NA were also summarized in S4 and S6 Tables.

5. Has the tMRCA been calculated?

Reply:

We very thank this reviewer for pointing out this issue. We did not calculate the tMRCA from our analyzed strains. But, according to your suggestions, we added this analysis in our revised manuscript.

In this revision, we utilized the MCMC algorithm of BEAST v1.10.4 to evaluate the tMRCA from our isolated strains. In detail, HKY substitution model and uncorrelated relaxed clock with lognormal relaxed distribution were used in our analysis, and the 10,000,000 MCMC chain length was performed. Furthermore, we used the TreeAnnotator v1.10.4 software of the BEAST v1.10.4 toolkit to annotate the target Maximum clade credibility tree (MCCTREE) with 10% Burnin as the number of trees. Please see our method section in the revised manuscript.

Additionally, we used FigTree v1.4.3 to visualize our MCCTREE. Divergence time of ancestral nodes was estimated with 95% highest density probability (HPD). Please see our revised manuscript.

6. HPD of nodes must be presented.

Reply:

Many thanks for this reviewer to help us point out this issue. We fully agree with your suggestions. According to your points, we calculated HPD for each node in our constructed MCCTREE using BEAST v1.10.4 and also demonstrated them in phylogeny using blue bars. And several focused ancestral nodes were highlighted for clade 6B.1A, 6B.1, 6B.2 and clade from all isolated strains by using BEAST v1.10.4 in this study. Please check our added Fig 4 in our revised manuscript.

For Mrbayes phylogeny for HA and NA, we further provided a Bayesian posterior probability range because of the algorithm without 95% HPD compared to MCCTREE. Please check our S1 and S3 Figs. 

Hope our revisions to meet your criterions. 

7. Genetic variation presented in the phylogeny has again been presented in the context of the HA antigenic sites and NA sites conferring neuraminidase resistance.

Reply: 

We thank the reviewer to point out this issue. We approved of your points. According to your suggestions, we have integrated mutation and phylogeny analysis into one section in our revised Results for this manuscript version. Please check our improvements accordingly.

8. The relevance of mutations belongs in the discussion but is presented in the results.

Reply: 

Thanks a lot. We have adjusted the relevance of mutations in Results to our Discussion.

9. THe prediction of “vaccine efficacy” is confusing to read. This section needs to be reworked to allow the reader to understand what terms like “worst-case” efficacy and “perfect match”mean. The table is confusing and must either be removed or presented in a more intelligible fashion.

Reply: 

Thanks for your important suggestions. The section of estimation of vaccine efficacy has been rewritten, and the unclear terms like “worst-case” efficacy and “perfect match” have been suppressed in this revision. The table about vaccine efficacy has been revised in a more intelligible mode. Please check our revisions in Results section <<Estimation of vaccine efficacy of influenza A(H1N1)pdm09 strains>>.

Discussion: The discussion does not highlight the relevance of the work. The discussion needs to be reworked to summarize key findings of the study and their agreement in a functional or epidemiological or evolutionary framework. Some of the content in the results belongs in the discussion e.g the paragraph on pg 19 “Several substitutions listed above…” etc.

Reply:

We thank the reviewer for pointing out this issue. We agree with your points. According to your suggestions, we have reworked to summarize our key findings in this revised manuscript. Please check our revisions in the Discussion section.

6. PLOS authors have the option to publish the peer review history of their article (what does this mean?). If published, this will include your full peer review and any attached files.

Do you want your identity to be public for this peer review? For information about this choice, including consent withdrawal, please see our Privacy Policy.

Reviewer #1: No

Reviewer #2: No

References

Deem MW, Pan K 2009. The epitope regions of H1-subtype influenza A, with application to vaccine efficacy. Protein Eng Des Sel 22: 543-546. 

Falchi A, Amoros JP, Arena C, Arrighi J, Casabianca F, Andreoletti L, Turbelin C, Flahault A, Blanchon T, Hanslik T, Varesi L 2011. Genetic structure of human A/H1N1 and A/H3N2 influenza virus on Corsica Island: phylogenetic analysis and vaccine strain match, 2006-2010. PLoS One 6: e24471. 

Gupta V, Earl DJ, Deem MW 2006. Quantifying influenza vaccine efficacy and antigenic distance. Vaccine 24: 3881-3888. 

Monamele CG, Munshili Njifon HL, Vernet MA, Njankouo MR, Kenmoe S, Yahaya AA, Deweerdt L, Nono R, Mbacham W, Anong DN, Akoachere JF, Njouom R 2019. Molecular characterization of influenza A(H1N1)pdm09 in Cameroon during the 2014-2016 influenza seasons. PLoS One 14: e0210119. 

Opanda S, Bulimo W, Gachara G, Ekuttan C, Amukoye E 2020. Assessing antigenic drift and phylogeny of influenza A (H1N1) pdm09 virus in Kenya using HA1 sub-unit of the hemagglutinin gene. PLoS One 15: e0228029. 

Pan K, Subieta KC, Deem MW 2011. A novel sequence-based antigenic distance measure for H1N1, with application to vaccine effectiveness and the selection of vaccine strains. Protein Eng Des Sel 24: 291-299. 

Tewawong N, Prachayangprecha S, Vichiwattana P, Korkong S, Klinfueng S, Vongpunsawad S, Thongmee T, Theamboonlers A, Poovorawan Y 2015. Assessing Antigenic Drift of Seasonal Influenza A(H3N2) and A(H1N1)pdm09 Viruses. PLoS One 10: e0139958.

---

## [Decision Letter · Decision Letter 1]

28 Apr 2020

PONE-D-20-01217R1

Whole-genome sequencing reveals origin and evolution of influenza A(H1N1)pdm09 viruses in Lincang, China, from 2014 to 2018

PLOS ONE

Dear Mrs Zhang,

Thank you for submitting your manuscript to PLOS ONE. After careful consideration, we feel that it has merit but does not fully meet PLOS ONE’s publication criteria as it currently stands. Therefore, we invite you to submit a revised version of the manuscript that addresses the points raised during the review process.

The revised manuscript is significantly improved. However, one of the experts  raised significant concern  (see below)  which need  to be corrected before the manuscript is suitable for publication. Please update the manuscript as advised by the Reviewer # 1 as soon as possible.

We would appreciate receiving your revised manuscript by Jun 12 2020 11:59PM. To enhance the reproducibility of your results, we recommend that if applicable you deposit your laboratory protocols in protocols.io, where a protocol can be assigned its own identifier (DOI) such that it can be cited independently in the future. For instructions see: http://journals.plos.org/plosone/s/submission-guidelines#loc-laboratory-protocols

We look forward to receiving your revised manuscript.

Kind regards,

Mrinmoy Sanyal, PhD

Academic Editor

PLOS ONE

Reviewers' comments:

Reviewer's Responses to Questions

**Comments to the Author**

1. If the authors have adequately addressed your comments raised in a previous round of review and you feel that this manuscript is now acceptable for publication, you may indicate that here to bypass the “Comments to the Author” section, enter your conflict of interest statement in the “Confidential to Editor” section, and submit your "Accept" recommendation.

Reviewer #1: (No Response)

2. Is the manuscript technically sound, and do the data support the conclusions?

Reviewer #1: Partly

3. Has the statistical analysis been performed appropriately and rigorously? 

Reviewer #1: Yes

4. Have the authors made all data underlying the findings in their manuscript fully available?

Reviewer #1: Yes

5. Is the manuscript presented in an intelligible fashion and written in standard English?

Reviewer #1: Yes

6. Review Comments to the Author

**Reviewer #1: **The authors have largely addressed my comments. There is one significant issue, though. The formula used for efficacy is the one for H3N2. But the authors are studying H1N1. So, the author should use the efficacy formula from from Ref. 16 Figure 2, which is

efficacy = 0.53 - 1.19 p_epitope

**All results should be updated with this formula.**

7. PLOS authors have the option to publish the peer review history of their article (what does this mean?). If published, this will include your full peer review and any attached files.

Reviewer #1: No

---

## [Author Response · Author response to Decision Letter 1]

1 Jun 2020

June 1st, 2020

To

The Editorial Office,

PloS One

Dear editors and reviewers:

Many thanks for receiving your kind reply. We are very grateful for the reviewer’s comments on our manuscript entitled “Whole-genome sequencing reveals origin and evolution of influenza A(H1N1)pdm09 viruses in Lincang, China, from 2014 to 2018” (PONE-D-20-01217R1). We have carefully studied your valuable comments especially for the formula calculating efficacy and tried our best to revise our manuscript. We believe that we have settled all of the concerns from the reviewer. We hope that our revisions can meet your approvals.

Kind regards,

Corresponding author: Mei-Ling Zhang and Yong Shao

E-mail address: meilingz2011@163.com

E-mail address: Yong_Shao_dws@126.com

Response to reviewers

Reviewers' comments:

Reviewer's Responses to Questions

Comments to the Author

1. If the authors have adequately addressed your comments raised in a previous round of review and you feel that this manuscript is now acceptable for publication, you may indicate that here to bypass the “Comments to the Author” section, enter your conflict of interest statement in the “Confidential to Editor” section, and submit your "Accept" recommendation.

Reviewer #1: (No Response)

2. Is the manuscript technically sound, and do the data support the conclusions?

Reviewer #1: Partly

Reply: According to suggestions from reviewer#1, we have fully corrected this issue in our revised manuscript.

3. Has the statistical analysis been performed appropriately and rigorously? 

Reviewer #1: Yes

4. Have the authors made all data underlying the findings in their manuscript fully available?

Reviewer #1: Yes

5. Is the manuscript presented in an intelligible fashion and written in standard English?

Reviewer #1: Yes

6. Review Comments to the Author

Reviewer #1: The authors have largely addressed my comments. There is one significant issue, though. The formula used for efficacy is the one for H3N2. But the authors are studying H1N1. So, the author should use the efficacy formula from from Ref. 16 Figure 2, which is

efficacy = 0.53 - 1.19 pepitope

All results should be updated with this formula.

Reply: We thank this reviewer to point out this issue for improving our manuscript. We really appreciate for your great professional knowledge in virology.

In this revision, according to your kind suggestions and your recommended reference and formula, we have updated all of our results throughout our manuscript including methods, results, and discussions. And we obtained a very similar efficacy trends with the previous formula. Therefore, all of revisions do not influence or change our conclusions.

Please check our revised manuscript in this version. We hope that our revisions can meet our requests sincerely.

7. PLOS authors have the option to publish the peer review history of their article (what does this mean?). If published, this will include your full peer review and any attached files.

Do you want your identity to be public for this peer review? For information about this choice, including consent withdrawal, please see our Privacy Policy.

Reviewer #1: No

---

## [Decision Letter · Decision Letter 2]

4 Jun 2020

Whole-genome sequencing reveals origin and evolution of influenza A(H1N1)pdm09 viruses in Lincang, China, from 2014 to 2018

PONE-D-20-01217R2

Dear Dr. Zhang,

We’re pleased to inform you that your manuscript has been judged scientifically suitable for publication and will be formally accepted for publication once it meets all outstanding technical requirements.

Kind regards,

Mrinmoy Sanyal, PhD

Academic Editor

PLOS ONE

Additional Editor Comments (optional):

Reviewers' comments:

Reviewer's Responses to Questions

**Comments to the Author**

1. If the authors have adequately addressed your comments raised in a previous round of review and you feel that this manuscript is now acceptable for publication, you may indicate that here to bypass the “Comments to the Author” section, enter your conflict of interest statement in the “Confidential to Editor” section, and submit your "Accept" recommendation.

Reviewer #1: All comments have been addressed

2. Is the manuscript technically sound, and do the data support the conclusions?

Reviewer #1: (No Response)

3. Has the statistical analysis been performed appropriately and rigorously? 

Reviewer #1: (No Response)

4. Have the authors made all data underlying the findings in their manuscript fully available?

Reviewer #1: (No Response)

5. Is the manuscript presented in an intelligible fashion and written in standard English?

Reviewer #1: (No Response)

6. Review Comments to the Author

Reviewer #1: The authors have addressed my comments. This is an interesting manuscript. It should be published.

7. PLOS authors have the option to publish the peer review history of their article (what does this mean?). If published, this will include your full peer review and any attached files.

Reviewer #1: No

---

## [Editor Report · Acceptance letter]

12 Jun 2020

PONE-D-20-01217R2 

Whole-genome sequencing reveals origin and evolution of influenza A(H1N1)pdm09 viruses in Lincang, China, from 2014 to 2018 

Dear Dr. Zhang:

I'm pleased to inform you that your manuscript has been deemed suitable for publication in PLOS ONE. Congratulations! Your manuscript is now with our production department. 

Kind regards, 

on behalf of

Dr. Mrinmoy Sanyal 

Academic Editor

PLOS ONE